# Latent Classes for the Treatment Outcomes in Women with Gambling Disorder and Buying/Shopping Disorder

**DOI:** 10.3390/jcm11133917

**Published:** 2022-07-05

**Authors:** Roser Granero, Fernando Fernández-Aranda, Milagros Lizbeth Lara-Huallipe, Mónica Gómez-Peña, Laura Moragas, Isabel Baenas, Astrid Müller, Matthias Brand, Claudia Sisquellas, Susana Jiménez-Murcia

**Affiliations:** 1Department of Psychobiology and Methodology, Universitat Autònoma de Barcelona-UAB, 08193 Bellaterra, Barcelona, Spain; roser.granero@uab.cat; 2Ciber Fisiopatología Obesidad y Nutrición (CIBERobn), Instituto Salud Carlos III, 28220 Majadahonda, Madrid, Spain; ffernandez@bellvitgehospital.cat (F.F.-A.); ibaenas@bellvitgehospital.cat (I.B.); 3Psychoneurobiology of Eating and Addictive Behaviors Group, Neuroscience Program, Institut d’Investigació Biomèdica de Bellvitge-IDIBELL, 08908 L’Hospitalet de Llobregat, Barcelona, Spain; 4Department of Psychiatry, Hospital Universitari de Bellvitge, 08907 L’Hospitalet de Llobregat, Barcelona, Spain; milagroslizbeth.lh@gmail.com (M.L.L.-H.); monicagomez@bellvitgehospital.cat (M.G.-P.); lmoragas@bellvitgehospital.cat (L.M.); csisquellas@bellvitgehospital.cat (C.S.); 5Department of Clinical Sciences, School of Medicine and Health Sciences, Universitat de Barcelona-UB, 08908 L’Hospitalet de Llobregat, Barcelona, Spain; 6Department of Psychosomatic Medicine and Psychotherapy, Hannover Medical School, 30625 Hannover, Germany; mueller.astrid@mh-hannover.de; 7General Psychology, Cognition and Center for Behavioral Addiction Research (CeBAR), University of Duisburg-Essen, 47057 Duisburg, Germany; matthias.brand@uni-due.de; 8Erwin L. Hahn Institute for Magnetic Resonance Imaging, 45141 Essen, Germany

**Keywords:** buying/shopping disorder, gambling disorder, women, cognitive behavioral therapy, latent class analysis

## Abstract

Background: The risk for behavioral addictions is rising among women within the general population and in clinical settings. However, few studies have assessed treatment effectiveness in females. The aim of this work was to explore latent empirical classes of women with gambling disorder (GD) and buying/shopping disorder (BSD) based on the treatment outcome, as well as to identify predictors of the different empirical groups considering the sociodemographic and clinical profiles at baseline. Method: A clinical sample of *n* = 318 women seeking treatment for GD (*n* = 221) or BSD (*n* = 97) participated. Age was between 21 to 77 years. Results: The four latent-classes solution was the optimal classification in the study. Latent class 1 (LT1, *good progression to recovery*) grouped patients with the best CBT outcomes (lowest risk of dropout and relapses), and it was characterized by the healthiest psychological state at baseline, the lowest scores in harm avoidance and self-transcendence, and the highest scores in reward dependence, persistence, self-directedness and cooperativeness. Latent classes 3 (LT3, *bad progression to drop-out*) and 4 (LT4, *bad progression to relapse*) grouped women with the youngest mean age, earliest onset of the addictive behaviors, and worst psychological functioning. Conclusions: GD and BSD are complex conditions with multiple interactive causes and impacts, which need wide and flexible treatment plans. Specific interventions should be designed according to the specific profiles of women for achieving early inclusion, retention and well-maintained long-term effects.

## 1. Introduction

Behavioral addictions refer to a form of addiction involving a compulsion to engage in a rewarding non-substance-related behavior, with concurrent maladaptative behaviors that lead to severe distress in diverse areas of the individuals’ functioning: reduced quality of life, family/social discord, comorbid physical/mental disorders, work impairment and financial problems [1,2,3,4]. During recent years, scientists and clinicians have focused attention on addictions without psychoactive substances, and the number of similarities revealed between drug addictions and non-substance addictions (in the form of addictive activity cravings, tolerance and abstinence syndrome, brain and nervous system correlates and bio-psychosocial consequences) pointed to the growing problem of these addictive behaviors within our communities [5].

GD is the only behavioral addiction included in the last version of the Diagnostic and Statistical Manual of Mental Disorders DSM-5 [6]. It has been incorporated in the broader category of “Substance-Addiction and Related Disorders” on the basis of cumulative empirical research suggesting an overlapping phenomenology, comorbidity and neurobiology with substance use disorders [7,8]. GD was also included in the latest revision of the International Classification of Diseases ICD-11 [9] as a disorder due to addictive behaviors. It is characterized by frequent concerns about gambling, gambling with larger amounts of money to receive the same level of desired experience (tolerance), repeated unsuccessful efforts to control or stop gambling, and restlessness or irritability when stopping gambling (abstinence). The most recent epidemiological studies estimate prevalence for GD close to 1% in industrialized countries across five continents [10], and considering joint problematic (non-disordered) gambling and GD, the prevalence rates increased to 6% assessing the last year of the survey, and to 10% for lifetime estimates [11].

The concept of behavioral addiction is relatively new in the field of psychopathology. In this category a wide variety of clinical conditions are grouped, and gambling disorder (GD) and buying/shopping disorder (BSD) are considered most prevalent in clinical and population-based samples. While no optimal nosological classification exists towards these mental disorders when considered together, prevailing suggestions as to the preferred approach include problematic gambling and problematic buying within the impulsive–compulsive spectrum [12,13]. This spectrum -refers to a number of disorders drawn from different diagnostic categories on the basis of comparisons of phenomenology (largely, the role of obsessive–compulsive–impulsive features), natural history, family history, biological markers and treatment response. Parallel classification schemes have conceptualized GD and BSD as a dimension across a broad range of the problem symptoms continuum (valid in both clinical and community samples), being at one end the lowest risk of recreational behavior, followed by problematic behavior, and being at the other end of the dimension the highest risk of addiction [14,15]. The position along the dimensional continuum is associated with gradual increases–decreases in the level of the addiction-related harms.

BSD is considered a behavioral addiction characterized by intense preoccupations with buying and possessing consumer goods that are not strictly necessary (patients buy more products than they can afford, and these products are neither frequently used or needed), leading to adverse consequences [16,17]. Proposals to include BSD within psychiatric taxonomies (such as DSM) have not been accepted, warranting further research [18,19]. The prevalence of BSD has been estimated within a large range between 1% to 8% in industrialized countries (this wide interval is related to the large heterogeneity of the study samples and the measurement tools), with a mean point estimate of 5% according to a meta-analysis [20]. Furthermore, both clinical and population-based studies have reported an increasing trend for BSD in developing consumer societies [21], highlighting the central relevance of materialistic values among these cultures as a predisposing factor to compulsive buying attitudes [22,23]. Epidemiological research has also reported an increasing propensity toward BSD among young adults and women, suggesting a greater tolerance for women and younger age individuals to make excess purchases [24]. Recent research also demonstrated that BSD constitutes a wide-ranging clinical condition for which sociodemographic features and personality traits have proven the capacity to discriminate between empirical clusters representative of distinctive clinical profiles [25]. The level of BSD symptoms is related to female gender, young age of onset of the addictive behavior, high levels in harm avoidance, low levels in self-directedness and high likelihood of comorbid psychopathology [26]. However, studies in this area are scarce, and some studies have failed to identify variables associated with the onset and progression of the BSD patients [27,28]. On the other hand, the scant consideration of assessment methods for impulsive buying has led to inconsistences in the research and has hampered cross-study comparability [29].

Individuals across a range of behavioral addictions show similar psychosocial and clinical patterns, both compared to control groups and as a function of the severity of the addictive behaviors. Considering the endophenotypes of GD and BSD, the role certain personality traits play in the onset and progression of the disorders, such as high levels of impulsivity (patients have a diminished control over impulses to engage in addictive behaviors) and high sensation seeking [30] have been observed to be similar. Certain sociodemographic variables have also been identified as potential risk factors, such as younger age (the age of onset is typically during young adulthood), disadvantaged social groups, and urban location (compared to rural location) [31,32]. The patients’ perceived motivations for the onset and progression of the disorders also appear similar for GD and BSD; individuals associate the addictive behavior episodes with pleasure and other positive feelings during the first stages, but the addictive episodes are increasingly used to alleviate negative moods when the condition progresses to impairment stages [33]. Other central features explaining the onset and the evolution of GD and BSD are deficient coping skills, low emotion regulation capacities [34,35] and implicit cognitive biases (such as difficulties evaluating long-term negative consequences and impairment in the capacity to money manage) [36,37]. Regarding neurobiological systems and neurocognitive characteristics, studies have shown common mechanisms compared with substance-related disorders. These include abnormalities in neurotransmitter systems (dopamine, serotonin and glutamate), disturbances in the motivation–reward systems and alterations in the reward-directed behavioral circuitry (primary ventral striatum and medial prefrontal cortex) [38,39,40,41,42,43,44]. Within this research area, the interaction of person–affect–cognition–execution (I-PACE) model has also been proposed for describing the psychological and neurological processes of problematic addictive behaviors, including GD and BSD [45]. Steep delay discounting (considered a measure of impulsivity, strongly related to the ability to delay gratification and described as a greater preference for smaller immediate rewards instead of larger delayed rewards) has also been defined as a common cognitive phenomenon for a range of addictions, including substance-related disorders and behavioral addictions [46]. Finally, an elevated risk for other comorbid psychiatric disorders is also characteristic of GD and BSD (these disorders rarely arise as a problem in isolation), being the most frequent concurrent conditions among females with anxiety disorders, depressive disorders and other problems within the impulse-control spectrum (substance use disorders, OCD, kleptomania, trichotillomania, bulimia or binge eating) [47,48,49,50,51]. The study of the comorbid presence of GD with BSD has also suggested the existence of underlying common etiological pathways for these two disorders [52,53].

Regarding intervention studies, evidence-based best practice studies suggest that cognitive behavioral therapy (CBT) constitutes a *gold standard* for many mental problems [54], with promising results for a broad range of addictive behaviors, including GD [55] and BSD [56]. CBT is a problem-solving approach centered on correcting the irrational thoughts associated with the addictive behaviors and their adverse consequences. The key objective of this intervention is to help subjects change the way they think and also the way they act. During this therapy, patients learn to identify and change cognitive biases and improve emotion regulation, since the modification of these thought patterns contributes to interrupting problem behaviors. Several cognitive and behavioral techniques are included in CBT programs, such as stimulus control, self-reinforcement, live exposure with response prevention, skills training and relapse prevention strategies (through other activities that are also rewarding and non-harmful). However, despite a significant body of literature assessing the efficacy of CBT for men diagnosed with GD, few treatment studies have focused on exploring the treatment outcomes for women with GD [57] and BSD [58]. This study is intended to provide new scientific evidence on the response to CBT in a clinical sample of women with behavioral addictions, specifically for the GD and the BSD subtypes. The results obtained in this work will allow the identification of latent groups of women with good and bad course trajectories, as well as predictive variables for the empirical latent classes.

### Objectives

The aims of this study were to explore latent classes of women with GD or BSD considering the CBT outcomes and to identify predictors of the different empirical classes. Based on previous studies, we hypothesized that different profiles characterize the progression of the behavioral addictions during the treatment of GD and BSD and that the baseline state will achieve discriminative capacity on the empirical latent classes.

## 2. Materials and Methods

### 2.1. Participants and Procedure

The sample in this work included *n* = 318 women consecutively attending the Pathological Gambling Unit and other Behavioral Addictions located in the Bellvitge University Hospital (Barcelona, Spain). Criteria for the study were adult age (18 years-old and older) and meeting clinical criteria for GD or BSD (according to different diagnostic measures, see below). Exclusion criteria were male sex and impairing neurological or psychiatric diseases, such as dementia, intellectual incapacity or active psychotic or bipolar episode, as determined by assessment with the tools used in the study. 

The presence of GD was identified in *n* = 221 women, while *n* = 97 presented BSD (no participant in the study presented the dual condition of GD + BSD). In the total sample, age was between 21 to 77 years (mean = 47.3, SD = 12.3). Mean age of onset of the behavioral addiction was 36.6 years (SD = 12.1), and the mean duration of the addiction-related problems was 5.7 years (SD = 5.4). Most women had achieved primary (54.1%) or secondary (35.8%) education levels, were married or lived with a stable partner (39.6%), were employed (50.6%) and belonged to mean-low or low socioeconomic levels (78.9%).

### 2.2. Assessment

*Diagnostic Questionnaire for Pathological Gambling* (according to DSM criteria) [59]. It was initially developed as a self-report diagnostic tool to identify the presence of GD through 19 items measuring the DSM criteria. It is currently used to assess both the diagnosis of GD based on the DSM-IV [60] and DSM-5 [6] taxonomies. The Spanish version used in this study has evidenced adequate psychometrical properties (Cronbach’s alpha α = 0.81 for a population-based sample and α = 0.77 for a clinical sample) [61]. In this work, this questionnaire was used to confirm the presence of GD in the subsample of women who seek treatment due to the gambling-related problems. The internal consistency achieved in this study was adequate (α = 0.79).

*Buying/Shopping Disorder Diagnosis* was assessed with a structured clinical face-to-face interview modeled after the SCID-I [62], developed to assess the presence of impulsive control disorders such as BSD [63,64]. The criteria used in this study have received wide acceptance in the research community [65], which must be considered due to the lack of diagnostic criteria for BSD in the most frequently used taxonomies (such as the DSM) and the recommendation of assessing the disorder through face-to-face interviews [66].

*Symptom Checklist-Revised* (SCL-90-R) [67]. This questionnaire was developed as a screening tool to assess the psychological state in multiple domains. It was planned as a self-report questionnaire, with 90 items structured into 9 primary (first order) dimensions (somatization, obsessive–compulsive, interpersonal sensitivity, depression, anxiety, hostility, phobic anxiety, paranoid ideation and psychoticism), and 3 global indices (global severity index (GSI), total positive symptoms (PST) and positive symptoms discomfort index (PSDI)). The Spanish adaptation of the SCL-90R used in this study has reported adequate psychometrical properties [68]. The internal consistency measured with Cronbach’s-alpha in our sample was in the range adequate to excellent: 0.92 for somatization, 0.89 for obsessive–compulsive, 0.88 for interpersonal sensitivity, 0.92 for depression, 0.91 for anxiety, 0.84 for hostility, 0.87 for phobic anxiety, 0.76 for paranoid ideation, 0.86 for psychoticism and 0.98 for the global indices.

*Temperament and Character Inventory-Revised* (TCI-R) [69]. This questionnaire was developed to obtain a measure of the personality traits according to Cloninger’s multidimensional model. It was planned as a self-report with 240 items factorized in 7 general factors (4 for temperament (novelty seeking, harm avoidance, reward dependence and persistence), and 3 for character (self-directedness, cooperation and self-transcendence)). The Spanish adaptation used in this work has obtained adequate psychometrical indices (mean Cronbach’s alpha in the good range, α = 0.87) [70]. The internal consistency in the sample of the study was in the range adequate to good: 0.71 for novelty seeking, 0.83 for harm avoidance, 0.77 for reward dependence, 0.85 for persistence, 0.84 for self-directedness, 0.77 for cooperation and 0.85 and self-transcendence.

*Other Variables.* The sociodemographics and the behavioral addiction-related variables were assessed with a semi-structured interview. This tool covered sex, marital status, education level, employment status and the socio-economic position index according to Hollingshead’s scale (this scale generated a classification based on the employment status, the participants’ level of education and the occupational prestige) [71]. Patients were also asked about the age of onset and the duration of the addiction-related problems, the cumulated debts due to the behavioral addiction and the presence of autolysis and suicidal ideation.

Data for the semi-structured interview were collected by psychologists and psychiatrists with high experience in the treatment of behavioral addictions. The clinicians also helped participants complete the self-report questionnaires (clarifying the meaning of possible items for patients to understand) to guarantee that no data were missing (for example, due to the lack of understanding).

### 2.3. CBT Program

The CBT program used in this study has been extensively described in previous studies [72]. The complete program was developed by qualified clinicians, expert in the application of this treatment among patients with behavioral addictions. The program was implemented through 12 weekly sessions lasting between 45 and 90 min each, in individual outpatient format in a the hospital unit setting. The key primary objective was to achieve full abstinence from all types of gambling or compulsive buying/shopping episodes. Techniques covered included cognitive restructuring, assertiveness training, self-reinforcement and stimulus control (the time for this concrete technique was flexible and determined on a case-by-case basis depending on the patients’ progress).

At the beginning of the CBT program, a psychoeducation session was aimed to: (a) provide knowledge about the concept of GD/BSD (as loss of control (addictive) disorders with several negative consequences on functioning); (b) provide information about the treatment (objectives, relevance to complete tasks and to remain in the intervention program, importance of total and permanent abstinence and possibility of sporadic relapses); and (c) collect diaries of baseline gambling or compulsive buying behaviors. During the next session of the treatment, the patients learned the CBT techniques (cognitive restructuring and problem-solving) and how to complete a self-monitoring diary in which to record their problematic behavior/s. Successive CBT sessions were focused on analyzing homework tasks, incidents, diaries, potential relapse/s, alternative activities carried out and attainment level with the treatment guidelines.

At the end of the CBT, patients and clinicians assessed the changes observed during the intervention and discussed expectations for the future (with regard to maintaining functional safe behaviors avoiding risk situations). Patients with GD are also encouraged on total and permanent abstinence, and patients with BSD on abstinence of compulsive buying episodes. During all the treatment, the presence of relapse (gambling or compulsive buying episodes) were registered, as well as the time/week of these events. For patients who dropout, the time of leaving the treatment was also registered.

Regarding the reliability of the CBT program employed in this research, previous studies have shown the short- and medium-term effectiveness, including samples of women meeting clinical criteria for GD [73,74] and BSD [75].

### 2.4. Statistical Analysis

This study used Latent Class Analysis (LCA) in Mplus8.1 [76] to explore the existence of empirical sub-groups among the complete sample based on the CBT outcomes. LCA is a classification procedure employed to relate a set of an observed dataset (including both categorical and quantitative variables) to a latent unobserved variable. This method is included within the measurement models in which individuals are classified into mutually exclusive/exhaustive types (named latent classes) based on the underlying patterns on a set of indicator variables. This study used the Robust Maximum Likelihood (MLR) estimator in the *Analysis* command, defined as indicators of the main treatment outcomes (dropout, time to dropout, relapses, time to relapses and euros lost in the relapses), and including the patients’ age, the diagnostic subtype (GD versus BSD), the duration of the behavioral addiction and the global psychopathological distress at baseline (SCL-90R GSI as covariates). The selection of the number of latent classes was based on the following criteria [77]: (a) the lowest Akaike (AIC) and Bayesian information criterion (BIC) indexes for the model (compared with other solutions); (b) entropy (measure of the model’s discriminative capacity, that is, its ability to identify individuals following the different latent classes) above 0.80; (c) high on-diagonal average values (above 0.80) in the matrix containing the probabilities of membership (that is, high average latent class probabilities for most likely latent class membership by latent class); (d) enough sample size in a class to allow for statistical comparisons; and (e) adequate clinical interpretability.

After obtaining the latent classes, these empirical groups were compared to the sociodemographics and the baseline clinical measures (registered previous to the CBT program), with the aim of identifying factors with discriminative capacity to the treatment outcomes. Comparisons were carried out with chi-square tests (χ^2^) for categorical variables and with analysis of variance (ANOVA) for quantitative measures. The estimate of the effect sizes for the proportion and the mean differences were based on standardized Cohen’s-*d* coefficients, considering poor-low effect size for |*d*| > 0.20, moderate-medium for |*d*| > 0.5 and large-high for |*d*| > 0.80 [78]. In addition, the increase in Type-I errors due to the multiple statistical tests for comparing latent classes was controlled with the Finner method, which constitutes a stepwise familywise error rate procedure [79].

Survival analysis was also used to describe the hazard rate to dropout and relapse and to compare the cumulative function between empirical latent classes. Survival procedures are statistical techniques used for modeling censored data, which occurs in longitudinal studies when patients withdraw from the study (that is, arrive at the end of the follow-up or is lost to the follow-up without event occurrence at the last measurement time) [80,81]. This study estimated the cumulative survival function with the Kaplan–Meier (product-limit) estimator, which provided the probability of women “living” during the CBT (in the work, survive is considered as the time without dropout or without the presence of relapse episodes). Comparison between the groups for the survival functions was done with the Log Rank (Mantel–Cox) test.

### 2.5. Ethics

The study was carried out in accordance with the Declaration of Helsinki principles, and approved by the Ethics Committee of University Hospital of Bellvitge (Ref. 307/06). All women provided signed informed consent for participating in the research. There was no financial or other compensation for being part of the sample of the study.

## 3. Results

### 3.1. Characteristics of the Sample

The first block of Table 1 displays the sociodemographics among GD and BSD patients, as well as the comparison between the groups. Compared to women with BSD, the proportion of women with lower education level and poorer social indexes was higher among women with GD. This diagnostic subtype also reported older chronological age and later age of onset of the behavioral problems (second block of Table 1). Regarding the psychopathology state and personality features at baseline, BSD was characterized by higher mean scores in the obsessive–compulsive dimension and novelty seeking trait (last block of Table 1).

### 3.2. Latent Classes

Table 2 includes the results of the classification process for the solutions based on latent classes 1 to 6. The most optimal solution selected in this study was the 4-latent class model (abbreviated as LT1, LT2, LT3 and LT4). This solution achieved the lowest AIC and BIC indexes, as well as the highest entropy value (compared to solutions for latent classes 2, 3 and 5). Solutions for latent classes 5 and 6 were not considered due the low sample size for some groups.

### 3.3. Comparison between the Latent Classes for the Treatment Outcomes

Table 3 displays the risk of dropout and relapse during the CBT. LT3 accumulated the highest incidence of dropouts and LT4 the highest incidence of relapse. LT1 included the lowest risk of relapse (as well as the lowest mean of euros spent in relapse episodes). Kaplan–Meier functions for the rate of dropout and relapse in the study are plotted in Figure 1 (LT1 was the group with the highest cumulative survival estimates for both these outcomes). Results of the Log Rank test achieved significant results for the cumulative survival curves for dropouts comparing LT1 versus LT3 (χ^2^ = 3.99, *p* = 0.046) and LT1 versus LT4 (χ^2^ = 5.48, *p* = 0.019). For the relapses, Log Rank tests obtained significant results comparing LT1 versus the other latent classes (χ^2^ = 9.99 and *p* = 0.002 compared to LT2, χ^2^ = 11.14 and *p* = 0.001 compared to LT3, and χ^2^ = 17.63, *p* < 0.001 compared to LT4). Based on the progression during the treatment, LT1 was labeled as “good progression to recovery”, LT2 as “middle progression”, LT3 as “bad progression to dropout” and LT4 as “bad progression to relapse”.

### 3.4. Comparison between the Latent Classes for Sociodemographic and Diagnosis Profile

The upper block of Table 4 shows the distribution of the diagnostic subtype within the latent classes (see also Figure 2). No statistical differences were found (effect size for the proportion differences were also within the low-poor range).

Table 4 also shows the distribution of the sociodemographic variables and the comparison between the latent classes. LT1 was characterized by including the highest proportion of patients with the highest education levels, employed and within the highest socioeconomic position indexes. LT2 was characterized by the highest proportion of patients with low education levels, unemployed and within the most unfavorable socioeconomic levels. No differences between LT3 and LT4 were found for the sociodemographic features.

### 3.5. Comparison between the Latent Classes for Clinical Measures at Baseline

LT1 was characterized by the healthiest psychopathological state (the lowest scores in the SCL-90R), the lowest scores in harm avoidance and self-transcendence and the highest scores in reward dependence, persistence, self-directedness and cooperativeness (see Table 5). LT2 included patients with the oldest mean age and the latest onset of the behavioral addiction, the lowest novelty seeking score and the highest self-transcendence level; this latent class also included the lowest proportion of patients who reported debts related to the behavioral addiction and autolysis behaviors. LT3 was defined by the youngest mean age, the earliest onset of the behavioral addiction, high scores in the psychopathological factors, high score in harm avoidance and the highest proportion of patients who reported autolysis behavior. LT4 also registered high scores in the psychopathology levels, high scores in harm avoidance and the lowest score in self-directedness.

## 4. Discussion

The present study aimed to explore the existence of latent classes in women with GD and BSD based on the CBT response, and to identify the variables with discriminant capacity in the empirical sub-groups identified in the LCA. The solution selected in this work as the optimal was the four latent classes, which achieved satisfactory fitting indexes and adequate clinical interpretation. 

The diagnostic subtype (GD/BSD) was statistically equally distributed between the latent classes, regardless of the differences in the baseline. In this study, GD women reported lower education levels and poorer social indexes than BSD women, while BSD women reported younger age, earlier onset of the disorder, higher levels in the obsessive–compulsive dimension and higher scores in novelty seeking than GD women. The absence of association between the diagnosis and the latent sub-groups is a relevant result which suggests that these two forms of behavioral addiction among women may benefit from CBT and obtain similar efficacy.

LT1 was the sub-group with the most efficacious treatment responses (good recovery with a very low risk of relapse and the lowest incidence rate of dropout). This class was characterized by including the highest prevalence of patients within the highest socioeconomic levels, the lowest level of comorbid psychological symptoms and the most functional personality profile (lower harm avoidance and self-transcendence and higher reward dependence, persistence, self-directedness and cooperativeness). These results are consistent with systematic reviews conducted within GD and BSD areas, which find that less psychopathology at intake is the most consistent predictor of success after treatment across multiple time points, followed by lower addictive behavior at the beginning of the interventions, higher education levels and more adaptive personality traits [82,83]; this evidence is applicable for both genders.

LT3 and LT4 were associated with the poorest CBT outcomes (highest incidence rates of dropout and relapses). These sub-groups clustered women with the worst psychopathology, most dysfunctional personality profile (the highest scores in novelty seeking and harm avoidance and lowest scores in persistence, self-directedness and reward-dependence) and the highest likelihood of autolysis behavior. The personality profiles grouped within LT3 and LT4 are characteristic of women with distance in interpersonal interactions, social withdrawal, low interest in pleasing others, passive avoidance behaviors, concern when anticipating potential danger/s, reduced responsibility for one’s own decisions, low self-esteem, lack of effectiveness to deal with daily situations and poor coping strategies. These attributes correlate with pessimistic behavior styles, the tendency towards shyness, frustration and to abandon goals at the slightest setback. Since these personality traits could contribute to poor treatment efficiency regardless of the behavioral addiction form, the severity of the addictive behaviors and the duration of the harms [58,84,85], these patients might benefit from the combination of CBT with other strategies. Such a strategy would increase motivation and make better use of treatment through motivational interventions (to improve the awareness of diseases), more flexible therapeutic guidelines, specific interventions focused on improving emotion regulation and goal setting other than definitive abstinence [86]. Patients with profiles defined by LT3 and Lt4 may even benefit from more intense treatment plans to encourage better attitudes and ensure that patients complete the follow-up and attain abstinence (for example, treatment plans with a larger number of sessions at increasing frequency). Carrying out the therapeutic strategies in group treatments (instead of individual programs, or a combination of individual and group treatments) may also feel more comfortable for patients characterized by LT3 and LT4 features. Knowing other women who share similar problems and struggles could increase interest for unmotivated patients and help them to hide their issues and avoid stigmatization/liability feelings. Even for those patients with more social difficulties, an Internet-based approach could represent an innovative and satisfactory format of CBT, enabling women with behavioral addictions to overcome many of the barriers related to conventional face-to-face formats [87].

LT3 was also defined by the youngest mean age, the earliest onset of the behavioral addiction and the longest duration of the disorder. Compared to LT3, LT4 clustered older women, later onset and shorter duration of the disorder. The mixed results regarding age, onset and duration suggest that the contribution of these variables on the treatment response interacts with other variables of the sociodemographic and clinical profiles [58,88]. Precise treatment protocols should consider the full specific profile of each patient, with the purpose of applying those techniques with the best research evidence.

Compared to the other latent classes, LT2 showed moderate CBT outcomes (worse compared to LT1 and better compared to LT3 and LT4). Specific characteristics of this cluster are the lower likelihood of debts related to the behavioral addiction, older chronological age, later onset of the behavioral addiction and shorter duration of the disorder. First, it is not surprising that less monetary expenditure related to the addictive behaviors was associated to the cluster grouping the oldest age patients and those individuals with the shortest duration of the disease, since this specific group had probably lower incomes and placed more moderate bets. However, previous studies have identified financial harm as a strong measure of the severity of the disease [89]; therefore, it should be supposed to be closely associated with poor treatment response. It must be noted that other studies have also observed that the patients’ sex interacts into the relationship between debts related to addictive behaviors and impairment levels. Moreover, it has been definitively shown that women problem gamblers with severe/extreme risks experience less significant financial consequences associated to addictive behaviors compared to women with a moderate risk [90,91]. Second, LT2 profile is representative of older individuals who exhibit a typical telescoping effect (addictive related problems develop more quickly than in younger age groups) [92] exacerbated by the typical aging-related cognitive biases [93]. Older individuals constitute a highly --vulnerable group with specific motivations for initiating and persisting in behavioral addictions (such as escaping loneliness and social isolation, relieving tension or coping with anxiety/depression symptoms due to the loss of a loved one or simply relieving boredom in retirement) [94,95]. Cognitive decline and physical–mental illness in older individuals also seem to play a central role in the onset, maintenance and escalation of addictive behaviors [96]. The absence of studies assessing the treatment response for behavioral addictions among older individuals does not allow us to know the specific role of these variables during the interventions. It seems essential that clinical settings adequately assess the concrete symptoms and negative consequences among older individuals with the aim of incorporating evidence-based integrative interventions to reduce physical and emotional problems. [97]. Healing-oriented holistic programs should include strategies to reduce chronic stress and impulsivity, and to improve social skills and emotional regulation (such as training in working memory and response inhibition). Medication should also be required in certain cases.

### Limitations

The evidence in this study has several limitations and additional questions for future research. First, the analyses were conducted on samples of women who met clinical criteria for GD and BSD, which limits the extrapolation of the findings to men and other behavioral addictions (like gaming disorder). Additionally, while we did not consider groups of women with an explicit comorbid substance-related disorders (or other psychiatric disorders), it is not clear whether our results could be also be valid for samples including dual pathology conditions.

## 5. Conclusions

To our knowledge, this is the first intervention study focused on exploring the existence of empirical latent classes of women seeking treatment for GD and BSD based on the CBT outcomes and aimed at identifying variables with discriminative capacity on the sub-groups. The identification of women-explicit features related to the treatment efficacy (that could be different from those reported for men) contribute to the knowledge of gender-specific processes involved in addictive behaviors and might be helpful for developing precise treatment plans for female patients The analysis of longitudinal data and multiple functioning areas is a further strength. Finally, the use of both person-centered and variable-centered methods constitutes an advantage. In longitudinal designs, person-centered methods are useful to identify sub-groups of individuals who share particular attributes and provide useful techniques for addressing questions concerning differences in patterns of progressions (for example assessing the course of a treatment through LCA). Variable-centered approaches complement the analysis, providing data on the association between variables and concretely addressing the relative contribution of some predictors (in this study for exploring the variables with discriminative capacity) on a concrete outcome.

## Figures and Tables

**Figure 1 jcm-11-03917-f001:**
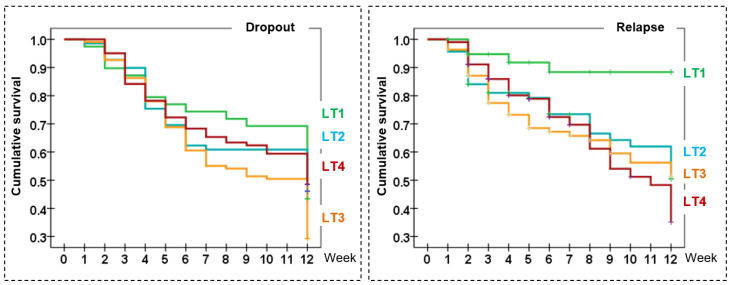
Kaplan–Meier functions for the incidence rate of dropout and relapse. Note. LT: latent class; LT1: good progression to recovery; LT2: middle progression; LT3: bad progression to dropout; LT4: bad progression to relapse.

**Figure 2 jcm-11-03917-f002:**
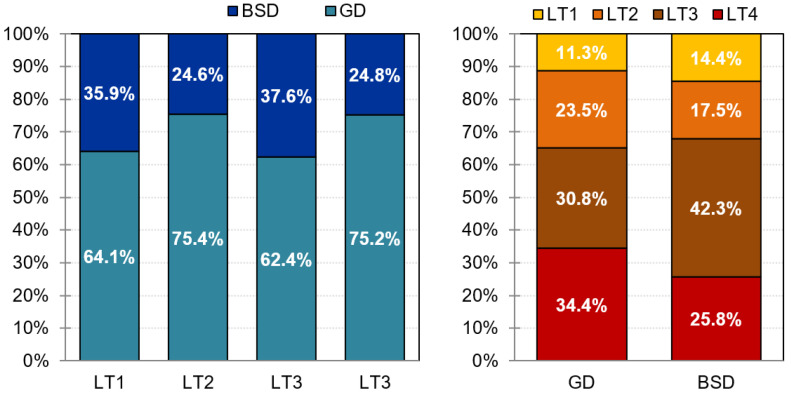
Distribution of the diagnostic subtype within each latent class. Note. BSD: buying/shopping disorder; GD: gambling disorder; LT: latent class; LT1: good progression to recovery; LT2: middle progression; LT3: bad progression to dropout; LT4: bad progression to relapse.

**Table 1 jcm-11-03917-t001:** Descriptive for the variables of the study.

	GD (*n* = 221)		BSD (*n* = 97)			
*Sociodemographic variables*	*n*	%	*n*	%	*p*	|*d*|
Education Primary	134	60.6%	38	39.2%	**0.001 ***	0.43
Secondary	76	34.4%	38	39.2%		0.10
University	11	5.0%	21	21.6%		**0.52 ^†^**
Marital status Single	95	43.0%	37	38.1%	0.649	0.10
Married/couple	84	38.0%	42	43.3%		0.11
Divorced/Separated	42	19.0%	18	18.6%		0.01
Employment Unemployed	110	49.8%	47	48.5%	0.828	0.03
Employed	111	50.2%	50	51.5%		0.03
Social Mean-high	10	4.5%	19	19.6%	**0.001 ***	**0.51 ^†^**
Mean	26	11.8%	12	12.4%		0.02
Mean-low	37	16.7%	26	26.8%		0.25
Low	148	67.0%	40	41.2%		**0.52 ^†^**
*Age-onset-duration*	*Mean*	*SD*	*Mean*	*SD*	*p*	*|d|*
Age (yrs-old)	49.14	12.28	43.22	11.44	**0.001 ***	**0.50 ^†^**
Onset of the addiction (yrs)	37.61	12.22	34.46	11.61	**0.032 ***	0.26
Duration of the addiction(yrs)	5.62	5.61	5.75	5.01	0.841	0.02
*Psychopathology(SCL-90R)*	*Mean*	*SD*	*Mean*	*SD*	*p*	*|d|*
Somatic	1.64	0.91	1.45	1.08	0.103	0.19
Obsessive–compulsive	1.59	0.84	1.82	1.02	**0.034 ***	0.25
Interpersonal sensitivity	1.45	0.87	1.50	1.02	0.670	0.05
Depressive	2.15	0.90	2.14	1.10	0.890	0.02
Anxiety	1.54	0.92	1.55	1.07	0.896	0.02
Hostility	1.10	0.84	1.30	1.00	0.066	0.22
Phobic	0.93	0.92	0.89	1.00	0.745	0.04
Paranoid	1.27	0.81	1.37	0.95	0.356	0.11
Psychotic	1.19	0.78	1.22	0.90	0.793	0.03
GSI score	1.54	0.74	1.56	0.89	0.838	0.02
PST score	56.81	18.46	54.79	21.42	0.393	0.10
PSDI score	2.28	0.59	2.36	0.70	0.308	0.12
*Personality (TCI-R)*	*Mean*	*SD*	*Mean*	*SD*	*p*	|*d*|
Novelty seeking	109.71	12.08	115.07	13.78	**0.001 ***	0.41
Harm avoidance	110.98	15.62	111.64	19.78	0.749	0.04
Reward dependence	101.14	13.06	103.53	16.36	0.168	0.16
Persistence	103.86	17.48	105.05	19.21	0.589	0.06
Self-directedness	117.96	18.09	121.43	22.77	0.147	0.17
Cooperativeness	133.01	13.25	134.32	15.84	0.445	0.09
Self-transcendence	68.29	15.15	66.89	16.67	0.460	0.09

Note. GD: gambling disorder; BSD: buying/shopping disorder; SD: standard deviation; * bold: significant comparison; ^†^ bold: effect size into the mild-moderate (|*d*| > 0.50) to large-high range (|*d*| > 0.80).

**Table 2 jcm-11-03917-t002:** Results of the LCA.

Model	Akaike	Bayesian	Sample-Size			Sample Size		Online
# Class.	(AIC)	(BIC)	Adjusted BIC	Entropy		Count	%	Probab.
1	39,372.954	39,508.388	39,394.203	1.000	T1	318	100.0%	1.000
2	20,854.429	20,986.101	20,875.089	0.959	T1	188	59.1%	0.993
					T2	130	40.9%	0.983
3	20,713.397	20,916.548	20,745.272	0.924	T1	44	13.8%	0.951
					T2	125	39.3%	0.972
					T3	149	46.9%	0.974
4	20,625.497	20,900.127	20,668.587	0.939	T1	39	12.3%	0.997
					T2	69	21.7%	0.912
					T3	109	34.3%	0.997
					T4	101	31.8%	0.963
5	20,762.98	21,109.089	20,817.285	0.895	T1	10	3.1%	0.973
					T2	40	12.6%	0.847
					T3	206	64.8%	0.962
					T4	52	16.4%	0.916
					T5	10	3.1%	0.979
6	20,591.177	21,008.764	20,656.696	0.894	T1	11	3.5%	0.957
					T2	33	10.4%	0.931
					T3	112	35.2%	0.953
					T4	100	31.4%	0.903
					T5	42	13.2%	0.945
					T6	20	6.3%	0.937

Note. # Class: number of latent classes; AIC: Akaike’s information criterion; BIC: Schwarz’s Bayesian criterion.

**Table 3 jcm-11-03917-t003:** Comparison between the latent classes for the CBT outcomes.

	LT1; *n =* 39			LT2; *n =* 69			LT3; *n =* 109			LT4; *n =* 101		
	*n*		*%*	*n*	*%*		*n*		*%*	*n*	*%*	
Risk of dropout	21		53.8%	39	56.5%		77		70.6%	52	51.5%	
Risk of relapse	4		10.3%	27	39.1%		43		39.4%	51	50.5%	
	*Mean*		*SD*	*Mean*	*SD*		*Mean*		*SD*	*Mean*	*SD*	
^1^ Number of sessions	9.26		3.04	8.41	2.90		8.03		2.83	8.59	2.93	
^1^ Number of relapses	0.41		0.76	0.87	1.28		0.97		1.38	1.63	2.07	
^1^ Euros spent/relapses	36.8		37.3	155.6	156.1		116.9		117.4	116.5	117.0	
	Pairwise comparisons											
	LT1–LT2		LT1–LT3		LT1–LT4		LT2–LT3		LT2–LT4		LT3–LT4	
	*p*	*|d|*	*p*	*|d|*	*p*	*|d|*	*p*	*|d|*	*p*	*|d|*	*p*	*|d|*
Risk of dropout	0.788	0.05	0.057	0.35	0.802	0.05	0.054	0.29	0.518	0.10	**0.004 ***	0.40
Risk of relapse	**0.001 ***	**0.70 ^†^**	**0.001 ***	**0.71 ^†^**	**0.001 ***	**0.93 ^†^**	0.966	0.01	0.144	0.23	0.108	0.22
	*p*	*|d|*	*p*	*|d|*	*p*	*|d|*	*p*	*|d|*	*p*	*|d|*	*p*	*|d|*
^1^ Number of sessions	0.156	0.29	**0.028 ***	0.42	0.243	0.22	0.392	0.13	0.679	0.06	0.155	0.20
^1^ Number of relapses	**0.019 ***	0.44	**0.002 ***	**0.50 ^†^**	**0.001 ***	**0.78 ^†^**	0.612	0.08	**0.003 ***	0.44	**0.007 ***	0.37
^1^ Euros spent/relapses	**0.001 ***	**1.05 ^†^**	**0.001 ***	**0.92 ^†^**	**0.001 ***	**0.92 ^†^**	0.077	0.28	0.077	0.28	0.981	0.00

Note. SD: standard deviation; ^1^ negative binomial model; * bold: significant comparison; ^†^ bold: effect size into the ranges mild-moderate to large-high; LT1: good progression to recovery; LT2: middle progression; LT3: bad progression to dropout; LT4: bad progression to relapse.

**Table 4 jcm-11-03917-t004:** Comparison between the latent classes for the diagnosis subtype and the sociodemographics.

	LT1; *n =* 39			LT2; *n =* 69			LT3; *n =* 109			LT4; *n =* 101		
	*n*		%	*n*	%		*n*		%	*n*	*%*	
Diagnosis GD	25		64.1%	52	75.4%		68		62.4%	76	75.2%	
BSD	14		35.9%	17	24.6%		41		37.6%	25	24.8%	
Education Prim.	14		35.9%	50	72.5%		60		55.0%	48	47.5%	
Secondary	18		46.2%	14	20.3%		40		36.7%	42	41.6%	
University	7		17.9%	5	7.2%		9		8.3%	11	10.9%	
Marital Single	12		30.8%	28	40.6%		52		47.7%	40	39.6%	
Married	19		48.7%	27	39.1%		41		37.6%	39	38.6%	
Divorced	8		20.5%	14	20.3%		16		14.7%	22	21.8%	
Employed Unempl.	9		23.1%	46	66.7%		46		42.2%	56	55.4%	
Employed	30		76.9%	23	33.3%		63		57.8%	45	44.6%	
Social Mean-high	7		17.9%	5	7.2%		10		9.2%	7	6.9%	
Mean	7		17.9%	5	7.2%		12		11.0%	14	13.9%	
Mean-low	8		20.5%	8	11.6%		29		26.6%	18	17.8%	
Low	17		43.6%	51	73.9%		58		53.2%	62	61.4%	
	Pairwise comparisons											
	LT1–LT2		LT1–LT3		LT1–LT4		LT2–LT3		LT2–LT4		LT3–LT4	
	*p*	*|d|*	*p*	*|d|*	*p*	*|d|*	*p*	*|d|*	*p*	*|d|*	*p*	*|d|*
Diagnosis GD	0.214	0.25	0.849	0.04	0.187	0.24	0.072	0.28	0.986	0.00	0.096	0.28
BSD												
Education Prim.	**0.001 ***	**0.75 ^†^**	0.072	0.39	0.352	0.24	0.053	0.36	**0.005 ***	**0.52 ^†^**	0.528	0.15
Secondary		**0.56 ^†^**		0.19		0.09		0.37		**0.47 ^†^**		0.10
University		0.33		0.29		0.20		0.04		0.13		0.09
Marital Single	0.553	0.21	0.184	0.35	0.521	0.19	0.524	0.14	0.973	0.02	0.323	0.16
Married		0.19		0.22		0.20		0.03		0.01		0.02
Divorced		0.01		0.15		0.03		0.15		0.04		0.18
Employed Unemplo.	**0.001 ***	**0.91 ^†^**	**0.034 ***	0.41	**0.001 ***	**0.68 ^†^**	**0.001 ***	**0.50 ^†^**	0.143	0.23	0.055	0.27
Employed												
Social Mean-high	**0.018 ***	0.33	0.266	0.26	0.144	0.34	**0.039 ***	0.07	0.306	0.01	0.378	0.08
Mean		0.33		0.20		0.11		0.13		0.22		0.09
Mean-low		0.25		0.14		0.07		0.39		0.18		0.21
Low		**0.63 ^†^**		0.19		0.36		0.43		0.27		0.17

Note. GD: gambling disorder; BSD: buying/shopping disorder; BA: behavioral addiction; SD: standard deviation; * bold: significant comparison; ^†^ bold: effect size into the ranges mild-moderate to large-high; LT1: good progression to recovery; LT2: middle progression; LT3: bad progression to dropout; LT4: bad progression to relapse.

**Table 5 jcm-11-03917-t005:** Comparison between the latent classes for the clinical measures at baseline.

	LT1; *n =* 39			LT2; *n =* 69			LT3; *n =* 109			LT4; *n =* 101		
	*Mean*		*SD*	*Mean*	*SD*		*Mean*	*SD*		*Mean*	*SD*	
Age (yrs)	44.69		8.45	61.51	7.05		35.87	7.50		51.04	7.79	
Onset of BA (yrs)	34.37		8.40	52.94	6.38		25.00	5.52		38.96	6.23	
Duration of BA (yrs)	5.21		4.58	4.12	3.70		7.08	6.60		5.36	5.01	
SCL-90R Somatic	0.64		0.65	1.40	0.80		1.72	1.02		1.92	0.87	
SCL-90R Obses.co.	0.66		0.56	1.35	0.73		1.91	0.84		2.00	0.83	
SCL-90R Sensitivity	0.42		0.47	1.22	0.69		1.74	0.89		1.75	0.89	
SCL-90R Depressive	0.91		0.79	1.88	0.79		2.43	0.80		2.50	0.85	
SCL-90R Anxiety	0.46		0.48	1.27	0.72		1.76	1.00		1.92	0.86	
SCL-90R Hostility	0.37		0.42	0.83	0.63		1.42	0.91		1.42	0.92	
SCL-90R Phobic	0.12		0.17	0.68	0.69		1.07	1.01		1.23	0.98	
SCL-90R Paranoid	0.49		0.46	1.03	0.62		1.47	0.85		1.62	0.85	
SCL-90R Psychotic	0.32		0.40	0.95	0.62		1.34	0.80		1.55	0.78	
SCL-90R GSI score	0.56		0.42	1.29	0.58		1.75	0.76		1.88	0.69	
SCL-90R PST score	27.90		15.53	50.43	15.41		61.74	16.80		65.08	13.64	
SCL-90R PSDI score	1.69		0.56	2.19	0.59		2.43	0.55		2.48	0.59	
TCI-R Novelty.se.	114.4		13.9	108.3	12.0		112.4	12.8		111.1	12.8	
TCI-R Harm avoid.	89.2		12.7	106.8	13.3		116.3	14.6		117.1	15.3	
TCI-R Reward dep.	109.6		12.5	102.0	13.1		102.2	15.7		98.4	12.5	
TCI-R Persistence	109.7		15.5	106.7	14.7		103.2	19.5		101.6	18.9	
TCI-R Self-directed.	142.4		17.2	122.5	13.4		116.0	19.3		110.9	17.0	
TCI-R Cooperative.	142.3		13.5	134.5	12.1		132.3	15.2		130.4	12.9	
TCI-R Self-Transcen.	62.8		17.3	72.2	13.3		66.5	16.6		68.4	14.7	
	*n*		%	*n*	%		*n*	%		*n*	%	
Debts due to BA	19		48.7%	20	29.0%		57	52.3%		47	46.5%	
Autolysis behavior	4		10.3%	4	5.8%		20	18.3%		14	13.9%	
Suicidal ideation	6		15.4%	15	21.7%		18	16.5%		22	21.8%	
	Pairwise comparisons											
	LT1–LT2		LT1–LT3		LT1–LT4		LT2–LT3		LT2–LT4		LT3–LT4	
	*p*	*|d|*	*p*	*|d|*	*p*	*|d|*	*p*	*|d|*	*p*	*|d|*	*p*	*|d|*
Age (yrs)	**0.001 ***	**2.16 ^†^**	**0.001 ***	**1.10 ^†^**	**0.001 ***	**0.78 ^†^**	**0.001 ***	**3.52 ^†^**	**0.001 ***	**1.41 ^†^**	**0.001 ***	**1.98 ^†^**
Onset of BA (yrs)	**0.001 ***	**2.49 ^†^**	**0.001 ***	**1.32 ^†^**	**0.001 ***	**0.62 ^†^**	**0.001 ***	**4.68 ^†^**	**0.001 ***	**2.22 ^†^**	**0.001 ***	**2.37 ^†^**
Duration of BA (yrs)	0.309	0.26	0.060	0.33	0.881	0.03	**0.001 ***	**0.55 ^†^**	0.138	0.28	**0.020 ***	0.29
SCL-90R Somatic	**0.001 ***	**1.05 ^†^**	**0.001 ***	**1.26 ^†^**	**0.001 ***	**1.67 ^†^**	**0.021 ***	0.35	**0.001 ***	**0.62 ^†^**	0.102	0.21
SCL-90R Obses.co.	**0.001 ***	**1.06 ^†^**	**0.001 ***	**1.74 ^†^**	**0.001 ***	**1.89 ^†^**	**0.001 ***	**0.70 ^†^**	**0.001 ***	**0.83 ^†^**	0.392	0.11
SCL-90R Sensitivity	**0.001 ***	**1.36 ^†^**	**0.001 ***	**1.87 ^†^**	**0.001 ***	**1.89 ^†^**	**0.001 ***	**0.66 ^†^**	**0.001 ***	**0.68 ^†^**	0.927	0.01
SCL-90R Depressive	**0.001 ***	**1.22 ^†^**	**0.001 ***	**1.91 ^†^**	**0.001 ***	**1.93 ^†^**	**0.001 ***	**0.70 ^†^**	**0.001 ***	**0.76 ^†^**	0.543	0.08
SCL-90R Anxiety	**0.001 ***	**1.33 ^†^**	**0.001 ***	**1.66 ^†^**	**0.001 ***	**2.11 ^†^**	**0.001 ***	**0.57 ^†^**	**0.001 ***	**0.83 ^†^**	0.167	0.17
SCL-90R Hostility	**0.005 ***	**0.85 ^†^**	**0.001 ***	**1.49 ^†^**	**0.001 ***	**1.47 ^†^**	**0.001 ***	**0.77 ^†^**	**0.001 ***	**0.76 ^†^**	0.978	0.00
SCL-90R Phobic	**0.002 ***	**1.12 ^†^**	**0.001 ***	**1.32 ^†^**	**0.001 ***	**1.57 ^†^**	**0.004 ***	0.45	**0.001 ***	**0.65 ^†^**	0.192	0.16
SCL-90R Paranoid	**0.001 ***	**0.98 ^†^**	**0.001 ***	**1.43 ^†^**	**0.001 ***	**1.65 ^†^**	**0.001 ***	**0.60 ^†^**	**0.001 ***	**0.80 ^†^**	0.163	0.17
SCL-90R Psychotic	**0.001 ***	**1.19 ^†^**	**0.001 ***	**1.62 ^†^**	**0.001 ***	**1.99 ^†^**	**0.001 ***	**0.56 ^†^**	**0.001 ***	**0.86 ^†^**	**0.037 ***	0.26
SCL-90R GSI score	**0.001 ***	**1.44 ^†^**	**0.001 ***	**1.92 ^†^**	**0.001 ***	**2.31 ^†^**	**0.001 ***	**0.67 ^†^**	**0.001 ***	**0.92 ^†^**	0.150	0.18
SCL-90R PST score	**0.001 ***	**1.46 ^†^**	**0.001 ***	**2.09 ^†^**	**0.001 ***	**2.54 ^†^**	**0.001 ***	**0.70 ^†^**	**0.001 ***	**1.01 ^†^**	0.118	0.22
SCL-90R PSDI score	**0.001 ***	**0.88 ^†^**	**0.001 ***	**1.35 ^†^**	**0.001 ***	**1.38 ^†^**	**0.007 ***	0.42	**0.001 ***	**0.50 ^†^**	0.507	0.09
TCI-R Novelty.se.	**0.017 ***	0.47	0.403	0.15	0.163	0.25	**0.035 ***	0.34	0.162	0.22	0.437	0.11
TCI-R Harm avoid.	**0.001 ***	**1.36 ^†^**	**0.001 ***	**1.98 ^†^**	**0.001 ***	**1.98 ^†^**	**0.001 ***	**0.68 ^†^**	**0.001 ***	**0.72 ^†^**	0.660	0.06
TCI-R Reward dep.	**0.007 ***	**0.59 ^†^**	**0.004 ***	**0.52 ^†^**	**0.001 ***	**0.89 ^†^**	0.944	0.01	0.094	0.28	**0.049 ***	0.27
TCI-R Persistence	0.397	0.20	0.052	0.37	**0.017 ***	0.47	0.207	0.20	0.071	0.30	0.523	0.08
TCI-R Self-directed.	**0.001 ***	**1.29 ^†^**	**0.001 ***	**1.44 ^†^**	**0.001 ***	**1.84 ^†^**	**0.014 ***	0.39	**0.001 ***	**0.76 ^†^**	**0.031 ***	0.28
TCI-R Cooperative.	**0.005 ***	**0.61 ^†^**	**0.001 ***	**0.70 ^†^**	**0.001 ***	**0.90 ^†^**	0.301	0.16	0.054	0.33	0.301	0.14
TCI-R Self-Transcen.	**0.002 ***	**0.61 ^†^**	0.201	0.22	0.054	0.35	**0.016 ***	0.38	0.114	0.27	0.366	0.12
	*p*	*|d|*	*p*	*|d|*	*p*	*|d|*	*p*	*|d|*	*p*	*|d|*	*p*	*|d|*
Debts due to BA	**0.040 ***	0.41	0.701	0.07	0.817	0.04	**0.002 ^†^**	**0.51 ***	**0.021 ^†^**	0.36	0.404	0.12
Autolysis behavior	0.395	0.17	0.239	0.23	0.568	0.11	**0.017 ^†^**	0.40	0.093	0.28	0.378	0.12
Suicidal ideation	0.423	0.16	0.870	0.03	0.396	0.17	0.382	0.13	0.995	0.00	0.331	0.13

Note. BA: behavioral addiction; SD: standard deviation; * bold: significant comparison; ^†^ bold: effect size into the ranges mild-moderate to large-high; LT1: good progression to recovery; LT2: middle progression; LT3: bad progression to dropout; LT4: bad progression to relapse.

## Data Availability

Data is available from the corresponding author.

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
