# Peer review of "Latent Classes for the Treatment Outcomes in Women with Gambling Disorder and Buying/Shopping Disorder"

_jcm, 2022, doi:10.3390/jcm11133917_

Round 1

Reviewer 1 Report

Dear authors,

Congratulations for the excellent paper. Very interesting and well thought out. Regarding the content I have nothing to criticize. Very well done. 

Regarding English language and editing I hardly found any mistakes in the first 3 parts: only in line 87 there are 2 points instead of 1 (".." instead of ".") at the end of a sentence.

But unfortunately the discussion has been written in poor English. Here some examples:

1. Line 412: "very few risk" (should be "very low risk"); and "of relapses" (should be "of relapse"); and I suppose you ment "lowest incidence rate" and not "slowest".

2. Line 413: "the most advantage socioeconomic levels" is poor English.

3. Line 436: "other strategies to incentive and make better use" is poor English.

4. Line 441: either "and ensure" or "..., ensuring" (without "and").

5. Line 443: "instead of individual ..." (erase "in")

6. Line 444: "may also feel more comfortable in patients", either "feel ... for" or "be ... for"

7. Line 482: "illness during elderly" ? What does that mean?

8. Lines 483 to 490: This sentence is much to long and hard to understand. Please, make at least 2 shorter sentences out of it. And "The absence ... does not allow ..."; and "does not allow to know" is poor English.

9. Line 493: "brain chemical imbalance" is very hard to define. If we would only give medication to does with proven "brain chemical imbalance", we would not be able to do our work. On the other hand, every addicted person must have some form of "brain chemical imbalance". Please, either erase this sentence, or write it differently. I suppose, you want to state, that medication can also be helpful in certain cases.

Author Response

Congratulations for the excellent paper. Very interesting and well thought out. Regarding the content I have nothing to criticize. Very well done. 

Response: On behalf of my co-authors, we greatly appreciate the Reviewer’s positive remarks and suggestions on our manuscript. We have considered all the comments and incorporated them into the revised submission. Changes to the original document are highlighted as red-color-font, and the itemized point-by-point response to the Reviewer is presented below.

Regarding English language and editing I hardly found any mistakes in the first 3 parts: only in line 87 there are 2 points instead of 1 (".." instead of ".") at the end of a sentence.

But unfortunately the discussion has been written in poor English. Here some examples:

  1. Line 412: "very few risk" (should be "very low risk"); and "of relapses" (should be "of relapse"); and I suppose you ment "lowest incidence rate" and not "slowest".
  2. Line 413: "the most advantage socioeconomic levels" is poor English.
  3. Line 436: "other strategies to incentive and make better use" is poor English.
  4. Line 441: either "and ensure" or "..., ensuring" (without "and").
  5. Line 443: "instead of individual ..." (erase "in")
  6. Line 444: "may also feel more comfortable in patients", either "feel ... for" or "be ... for"
  7. Line 482: "illness during elderly" ? What does that mean?
  8. Lines 483 to 490: This sentence is much to long and hard to understand. Please, make at least 2 shorter sentences out of it. And "The absence ... doesnot allow ..."; and "does not allow to know" is poor English.
  9. Line 493: "brain chemical imbalance" is very hard to define. If we would only give medication to does with proven "brain chemical imbalance", we would not be able to do our work. On the other hand, every addicted person must have some form of "brain chemical imbalance". Please, either erase this sentence, or write it differently. I suppose, you want to state, that medication can also be helpful in certain cases.

Response: We appreciate this comment. The reviewed version of the manuscript has been now revised by a professor of English Language, who is a professional translator/corrector. All the typos indicated by the Reviewer have been corrected, as well as other grammatical errors.

Reviewer 2 Report

The present paper aims at exploring latent empirical classes of a female sample (N=318) with gambling disorder and buying/shopping disorder based on the outcome to a CBT therapy. Another purpose of the research was to identify predicts of the different empirical groups based on baseline sociodemographic and clinical characteristics. 

The paper is interesting, well written, and clearly conducted. I have, therefore, only minor concerns and suggestions that I listed below:

- The introduction provide a clear rationale for the study. However, I suggest to the authors to better explain the choice of specifically including a female sample, since this aspect is not fully clear to me after reading the first part of the paper. 

- The authors stated that patients with psychotic or bipolar disorder were excluded. However, no information are present about the presence of patients with comorbid psychiatric disorders and/or taking drug therapy during the study. I suggest to add some information about this, if present. 

- A typo in line 175 is present: “In the totals ssample”.

- Table 1 is difficult to read. I suggest its structure or to split it in two different tables. 

Author Response

The present paper aims at exploring latent empirical classes of a female sample (N=318) with gambling disorder and buying/shopping disorder based on the outcome to a CBT therapy. Another purpose of the research was to identify predicts of the different empirical groups based on baseline sociodemographic and clinical characteristics. The paper is interesting, well written, and clearly conducted. I have, therefore, only minor concerns and suggestions that I listed below.

Response: On behalf of my co-authors, we would like to thank the Reviewer for your time, efforts and input into our manuscript. Please, find below our responses and edits. Changes to the original document are highlighted as red-color-font. We believe that you made this study much robust and clearer to contribute to the literature. Thank you.

- The introduction provide a clear rationale for the study. However, I suggest to the authors to better explain the choice of specifically including a female sample, since this aspect is not fully clear to me after reading the first part of the paper.

Response: We appreciate this thoughtful feedback provided from Reviewer. We have now explained-justified the study of the treatment outcomes among women with GD and CBT in the paragraph before the objectives:

“But despite a significant body of literature assessing the efficacy of CBT for men diagnosed of GD, few treatment studies have been focused on exploring the treatment outcomes for women with GD 57 and BSD 58. This study is intended to provide new scientific evidence on the response to CBT in a clinical sample of women with behavioral addictions, concretely for the GD and the BSD subtypes. The results obtained in this work will allow to identify latent groups of women with good and bad course-trajectories, as well as predictive variables for the empirical latent classes”.

- The authors stated that patients with psychotic or bipolar disorder were excluded. However, no information are present about the presence of patients with comorbid psychiatric disorders and/or taking drug therapy during the study. I suggest to add some information about this, if present. 

Response: Thank you for pointing this out. The psychological state was assessed in the study with the SCL-90R questionnaire, which measures the symptom-severity in nine primary-factors (somatization, obsessive-compulsive, interpersonal sensitivity, depression, anxiety, hostility, phobic anxiety, paranoid ideation, and psychotic ideation), and three global indexes measuring the psychological distress (GSI, PST and PSDI). The presence of any comorbid disorder was not assessed through standardized diagnostic tool, and therefore this information was not included in the study. Similarly, no information is available on the use of complementary pharmacological treatments. Since the reviewer's assessment seemed extremely relevant to us, we had included this text in the limitations subsection:

“Additionally, while we did not considered groups of women with an explicit comorbid substance-related disorder (or other psychiatric disorder), it is not clear whether our results could be also valid for samples including dual pathology conditions”.

- A typo in line 175 is present: “In the totals ssample”.

Response: Thank you for pointing this out. We have corrected this typo.

- Table 1 is difficult to read. I suggest its structure or to split it in two different tables. 

Response: We appreciate this helpful suggestion. We have re-ordered and re-structured the contents of this table.